# Refined Quantification of Infection Bottlenecks and Pathogen Dissemination with STAMPR

Karthik Hullahalli,[a,b] Justin R. Pritchard,[c,d] Matthew K. Waldor[a,b]

aDepartment of Microbiology, Harvard Medical School, Boston, Massachusetts, USA
bDivision of Infectious Diseases, Brigham & Women's Hospital, Boston, Massachusetts, USA
cDepartment of Biomedical Engineering, Penn State University, University Park, Pennsylvania, USA
dHuck Institute for Life Sciences, University Park, Pennsylvania, USA

**ABSTRACT** Pathogen population dynamics during infection are critical determinants of infection susceptibility and define patterns of dissemination. However, deciphering these dynamics, particularly founding population sizes in host organs and patterns of dissemination between organs, is difficult because measuring bacterial burden alone is insufficient to observe these patterns. Introduction of allelic diversity into otherwise identical bacteria using DNA barcodes enables sequencing-based measurements of these parameters, in a method known as STAMP (Sequence Tag-based Analysis of Microbial Populations). However, bacteria often undergo unequal expansion within host organs, resulting in marked differences in the frequencies of barcodes in input and output libraries. Here, we show that these differences confound STAMP-based analyses of founding population sizes and dissemination patterns. We present STAMPR, a successor to STAMP, which accounts for such population expansions. Using data from systemic infection of barcoded extraintestinal pathogenic *E. coli*, we show that this new framework, along with the metrics it yields, enhances the fidelity of measurements of bottlenecks and dissemination patterns. STAMPR was also validated on an independent barcoded *Pseudomonas aeruginosa* data set, uncovering new patterns of dissemination within the data. This framework (available at https://github.com/hullahalli/stampr_rtisan), when coupled with barcoded data sets, enables a more complete assessment of within-host bacterial population dynamics.

**IMPORTANCE** Barcoded bacteria are often employed to monitor pathogen population dynamics during infection. The accuracy of these measurements is diminished by unequal bacterial expansion rates. Here, we develop computational tools to circumvent this limitation and establish additional metrics that collectively enhance the fidelity of measuring within-host pathogen founding population sizes and dissemination patterns. These new tools will benefit future studies of the dynamics of pathogens and symbionts within their respective hosts and may have additional barcode-based applications beyond host-microbe interactions.

**KEYWORDS** barcodes, bottlenecks, population dynamics

During infection, microbial pathogens encounter a variety of barriers that impede colonization and help prevent subsequent disease. These obstacles include innate and adaptive effectors of the immune system, the microbiota, and anatomical and chemical barriers, such as stomach acidity and physical niche availability (1). Collectively, these restrictions, which generally act to protect the host and reduce the size of the pathogen population postinoculation, are often referred to as a "bottleneck." When bacteria in the inoculum contain multiple alleles, the allelic composition of the bacterial population found at sites of colonization will differ from that in the

Address correspondence to Karthik Hullahalli, hullahalli@g.harvard.edu, or Matthew K. Waldor, mwaldor@research.bwh.harvard.edu.

inoculum after passing through the bottleneck, a phenomenon more broadly referred to as genetic drift (2). In infection biology, bottlenecks are key determinants of whether a host becomes colonized by a pathogen, govern paths of dissemination within individual hosts, and influence transmission between hosts (3–9). However, the set of host mechanisms that govern bottlenecks remain incompletely understood. Genome-scale genetic screens in bacteria can be used to investigate host defense axes, but are themselves confounded by bottleneck effects that cause mutant strains in a population to be eliminated by the host by chance alone, rather than through selection (10–16).

Infection bottlenecks are difficult to quantify if the experimental inoculum is composed of bacteria of uniform genotype. Several methods that introduce allelic diversity have been used to circumvent this issue and measure bottlenecks (7, 17). One approach involves the introduction of artificial and fitness-neutral short random sequence tags (barcodes) into otherwise identical cells. The comparison of barcode abundances before and after infection through high-throughput DNA sequencing then enables bottleneck quantification. Combining barcoding with deep sequencing is widely generalizable and can be applied in several different contexts, such as experimental evolution and cancer progression (18). Different analytical approaches and metrics for comparisons of barcode frequencies have been created (19–21), including Sequence Tag-based Analysis of Microbial Populations (STAMP), for analysis of infection bottlenecks (22). In STAMP, deep sequencing is used to determine the distribution of barcode frequencies in an inoculum (the input) and in various organs (the output). The changes in barcode distributions (i.e., allele frequencies) between input and output are used to quantify the magnitude of genetic drift, which approximates the magnitude of the bottleneck (23). Bottlenecks are measured as the size of the founding population (FP), i.e., the number of unique cells from the inoculum that give rise to the population in a sample. A small FP value is indicative of a "tight" bottleneck, whereas a large FP indicates a "wide" bottleneck. FP is estimated by an application of an equation from Krimbas and Tsakas, originally used to quantify genetic drift in insect populations (23). In STAMP, the estimate of FP is known as $N_b$. We distinguish FP and $N_b$ to emphasize that FP is impossible to measure precisely, as it would require every cell in the inoculum to possess a different tag and infinite sequencing depth. $N_b$ calculation circumvents these limitations by quantifying the differences in barcode frequencies between a reference inoculum and output organ samples. STAMP has been used in several infection models across multiple anatomical sites to estimate FP and unveil host determinants of infection bottlenecks (22, 24–27). Recent work has also enabled the use of STAMP to measure bacterial replication and death rates (28).

Two key assumptions that underlie the calculation of $N_b$ are that (i) all sampled bacteria in the population have experienced a singular, identical bottleneck and that (ii) all cells grow at similar rates after passing through the bottleneck (22, 28). These assumptions oversimplify conditions within the host, as bacteria within an organ are likely exposed to different environments depending on their suborgan localization. For example, variation in the immune state of diverse host cells can impose different pressures on bacteria (29). Furthermore, organ reseeding events can result in multiple populations of bacteria within an organ that have undergone distinct bottlenecks. Phenotypic heterogeneity in the pathogen population can also influence post-bottleneck expansion rates (30, 31). These additional sources of variation in barcode frequencies result in a consistent underestimation of FP by $N_b$, because calculation of $N_b$ relies on comparing the similarity of barcode frequencies between an output organ sample and a diverse input. The $N_b$ value of an output sample will be larger if the barcode frequency distribution in the organ sample more closely resembles the inoculum. As genetic drift or uneven growth rates cause the barcode frequency distribution in the output sample to vary, $N_b$ decreases. However, $N_b$ alone cannot distinguish between genetic drift or uneven growth, and since both are prevalent in biological data, additional metrics are warranted and would markedly improve data interpretation. For example, two organs may possess very similar FPs but one organ may be permissive to

increased replication of a subpopulation. These organs would have different $N_b$ and would therefore be interpreted to differ in FP.

We found that in biological data, uneven growth often manifests as the expansion of very few clones, which are evident as disproportionately abundant barcodes and lead to consistent underestimates of the true FP by $N_b$. In infection contexts, uneven pathogen growth may arise from multiple causes, including local host permissiveness or phenotypic heterogeneity in the pathogen. Disproportionately abundant barcodes may suggest multiple distinct populations within an organ, but uneven growth may be present even within a single population. Here, we present STAMPR, a computational approach that overcomes these limitations of STAMP. STAMPR is a successor to STAMP that relies on an iterative barcode removal algorithm to account for the contribution of clonal expansion to bottlenecks. In addition, STAMPR employs additional metrics to evaluate dissemination patterns that characterize the extent to which individual barcodes contribute to bacterial spread. Using data from systemic infection of barcoded extraintestinal pathogenic *E. coli* (ExPEC) (32), we show that STAMPR enhances the fidelity of measurements of bottlenecks and dissemination patterns by accounting for every barcode. We use these tools to reanalyze an independently generated and published data set that explored *Pseudomonas aeruginosa* systemic spread (24). Our tools readily detected and quantified previously unappreciated instances of clonal expansion and dissemination in these data. STAMPR (freely available at https://github.com/hullahalli/stampr_rtisan) therefore enables a deeper and more complete understanding of within-host bacterial population dynamics.

## RESULTS AND DISCUSSION

**Highly abundant barcodes confound measurement of founding population sizes.** Our motivation for questioning the fidelity of $N_b$ as a proxy for FP came from observations where $N_b$ values were often much smaller than the number of detected barcodes in sequencing data. This discrepancy became particularly clear in analyses of STAMP-based experiments investigating within-host ExPEC dissemination, the biological findings of which are described further in a companion manuscript (32). In the ExPEC systemic infection model, the pathogen is inoculated intravenously and samples are taken from different organs to monitor dissemination and expansion. In multiple organs, we found clonally expanded bacterial populations, intermixed with less abundant, more diverse bacterial populations. By introducing additional variance to output barcode frequencies, these highly abundant "outliers" confounded $N_b$, as samples with hundreds of detectable tags yielded much lower $N_b$ values (occasionally >10 fold). The discrepancy between $N_b$ (as a true measure of FP) and the number of barcodes is not biologically plausible; if 100 barcodes are detected, the founding population must be composed of at least 100 unique bacteria. While it is possible for an individual cell to possess two barcodes, it is highly unlikely. During library preparation, individual colonies are Sanger sequenced, confirming that the presence of multiple tags per cell is below detection. Furthermore, within-run sequencing controls (samples with known numbers of barcodes) serve to rule out that cross contamination or sequencing errors significantly influence the data.

We sought to develop a computational approach that can recognize and account for disproportionately abundant barcodes. This approach would not only need to account for highly abundant tags, but be sufficiently unbiased to enable determination of FP when it is difficult to identify "outliers" by visual inspection of barcode frequency graphs. In an ideal system, every bacterium from the inoculum would be tagged with a single unique barcode, in which case counting the number of barcodes would yield a more accurate measure of FP than $N_b$. However, creation of highly diverse libraries has been technically challenging, particularly for non-model organisms. An alternative approach to improve the accuracy of FP estimates leverages the power of computational resampling, which, unlike $N_b$, is not affected by unequal growth rates. Simulations can be performed on the input at a variety of sampling depths to determine the sample depth $N_s$ that yields the same number of barcodes detected in the

output sample. For example, if 100 barcodes are detected from an output sample derived from an input library of 1,000 barcodes, then $N_s$ represents the number of reads that were sampled from the input such that 100 barcodes are detected; this value will always be slightly larger than 100. Therefore, $N_s$, unlike $N_b$, is not skewed when there is increased variation in barcode frequencies between an output organ and the input.

To demonstrate our methodology, we first artificially recreated a sample in which $N_b$ underestimates FP by using a series of barcode frequency distributions from known bottleneck sizes collected from *in vitro*-generated bottlenecks where FP sizes are known. Combining the barcode frequencies observed in an $\sim 1.4 \times 10^3$ CFU FP and an $\sim 1.4 \times 10^1$ CFU FP (Fig. 1A) yielded distributions as shown in Fig. 1B. This mimics a sample where $\sim 14$ cells have expanded faster than the other $\sim 1,400$ after both populations have passed through the same, singular bottleneck. The true FP for this artificial population is close to $1.4 \times 10^3$. However, the calculated $N_b$ is 150, $\sim 9$-fold lower than the true FP, because the expanded population is viewed in STAMP calculations as substantial variation between input and output barcode frequencies, leading to a marked underestimate of FP. Experimental data from the ExPEC model revealed similar patterns, where calculated $N_b$ values were lower than the number of detected barcodes, therefore smaller than the true FP (Fig. 2).

We developed an algorithm that provides a more complete estimate of FP (Fig. 1B to D, computational workflow described in the Materials and Methods section, and Text S1 in the supplemental material). Our approach was developed on computational samples (such as in Fig. 1A) and our ExPEC experimental data sets, and yields more accurate estimates of FP. In brief, the algorithm iteratively removes barcodes from the output sample (from greatest to least abundant) and calculates $N_b$ after each iteration. A better estimation of FP for the artificial sample described above is equal to the $N_b$ after the first $\sim 10$ most abundant barcodes are removed, which is $\sim 10^3$. Subsequent removal of barcodes does little to change $N_b$ (i.e., the y values plateau), and we refer to $\sim 10^3$ as a more "resilient" estimate of FP. We refer to plots of $N_b$ versus iteration as "resiliency plots" (Fig. 1C) and this algorithm as the "resiliency algorithm." The resiliency plot can be used to define "breaks" that delineate discrete subpopulations within the sample (shown as red lines in Fig. 1C and D, separating high-abundance barcodes, low-abundance barcodes, and noise). Then, these subpopulations are weighted by the fractional abundance of barcodes within breaks, enabling determination of a noise threshold. Whenever samples are multiplexed, index hopping results in noise, where usually <1% of reads are technical artifacts. Importantly, for most samples, noise represents a discrete subpopulation that can be detected by the resiliency algorithm. Removing noise is important because, in some cases, noise can comprise more barcodes than the true FP (e.g., Fig. 1A, sample with FP = 14).

After removing noise, a second resiliency plot is generated. Using this graph, the algorithm then determines the maximum possible value for $N_b$. In addition, the number of remaining barcodes is used to calculate $N_s$. The final output of the resiliency algorithm is referred to as $N_r$, which is set equal to the maximum value among (i) $N_s$, (ii) the initial $N_b$ estimate, or (iii) the new maximum $N_b$ from the second resiliency plot. $N_s$ is used in this manner since it is completely independent of relative barcode abundances and considers only their presence or absence. For example, populations that have undergone significant uneven growth post-bottleneck will have low $N_b$ values, so $N_s$ would yield the greatest estimate of FP. Furthermore, this logic ensures that $N_r$ will always be equal to or greater than $N_b$. By accounting for the presence of every barcode, this approach more accurately estimates FP regardless of the presence of disproportionately abundant barcodes in biological data (e.g., Fig. 2, Fig. S1). The sensitivity of $N_b$ to highly abundant tags can further be exploited by measuring the ratio of $N_r/N_b$, which in effect quantifies unequal barcode distributions and can provide information about clonal expansion.

Computationally combining additional samples (as in Fig. 1) further confirms that $N_r$ provides a more accurate assessment of FP than $N_b$ across most composite barcode

   

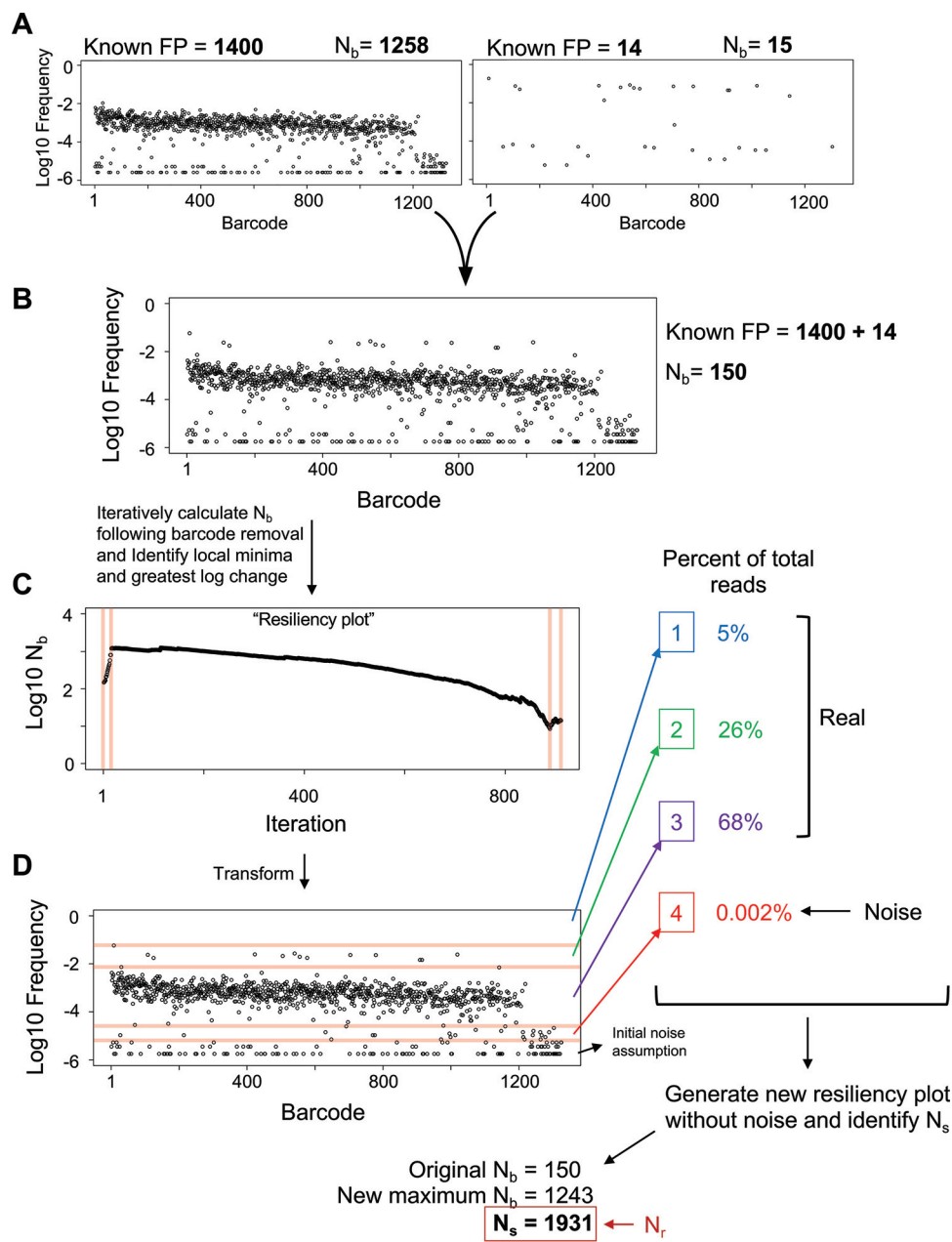

**FIG 1** Overview of the resiliency algorithm. (A) Two barcode frequency distributions from known CFU dilutions (and thus known FPs) are shown, with calculated $N_b$ values. The raw read counts of each barcode in these samples are summed to give rise to the plot in B. This results in a computational sample that mimics a scenario where ~14 cells have expanded at a much faster rate than the other ~1,400, despite both populations having undergone the same bottleneck. From the data plotted in B, the resiliency algorithm is run to generate a resiliency plot (displayed in C) by iteratively removing the most abundant barcodes and calculating $N_b$ after each iteration. Breaks in the graph (red lines in C) are algorithmically identified and visually superimposed on the plot in B, resulting in the diagram shown in D. Breaks signify potentially discrete populations or noise, and the weight of each population is determined as a fraction of total reads. An initial noise assumption is set by the user and, in this example, was set to 1% of all reads. The largest log change in the weights determines a computationally defined threshold for noise. From this, a new resiliency plot is generated without noise and the greatest value among the original $N_b$ value, the new maximum $N_b$ from the second resiliency plot, or $N_s$ is used to determine $N_r$. In this example, $N_s$ was the largest of these values, and is therefore the resulting value of $N_r$.

distributions (Fig. S2). While $N_b$ is accurate for single populations generated from *in vitro* standards, it fails to accurately calculate FP for composite populations, which more closely resemble *in vivo* data. However, since several parameters could potentially influence the output of the resiliency algorithm, we additionally conducted a

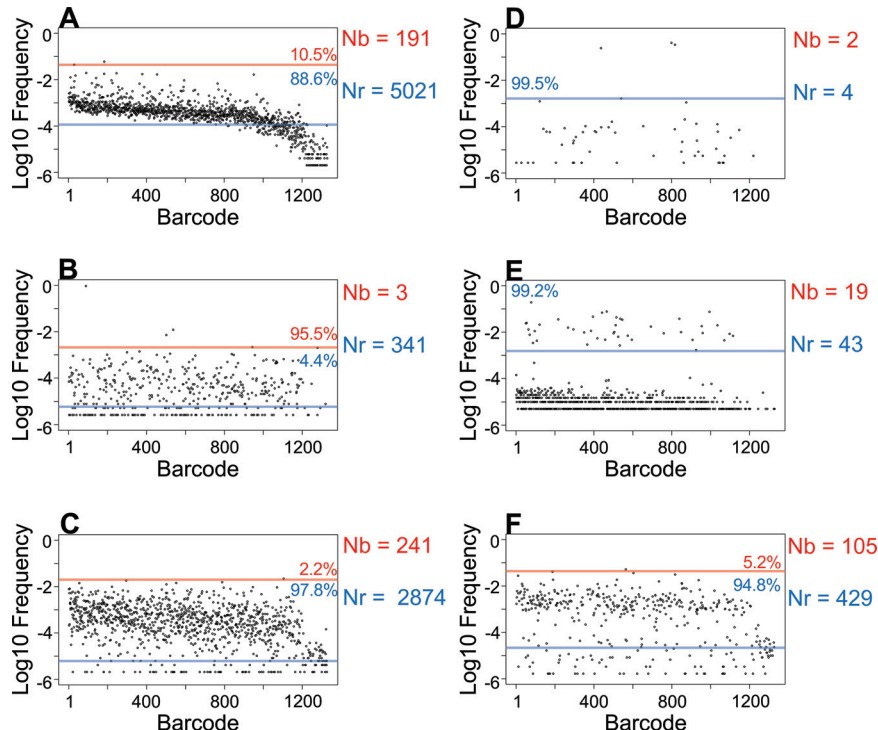

**FIG 2** Variations in $N_r$ and $N_b$. Barcode frequency distributions for six samples from ExPEC systemic infection (32) are shown. The *y* axes are displayed on a log scale to facilitate identification of noise. For each sample, a red line indicates the first break identified by the resiliency algorithm and delineates discrete subpopulations. The blue line indicates the threshold for noise. In each subpopulation, the fraction of total barcodes represented is displayed. $N_b$ and $N_r$ values are indicated.

series of simulations on computational samples modeled after the skew of the ExPEC and *P. aeruginosa* libraries (Fig. S3). When a single population is present, increasing the variability in growth rates after a single uniform bottleneck leads to a large decrease in $N_b$, while $N_r$ remains substantially closer to the true FP (Fig. S4A). In a similar manner, when a small subpopulation possesses a faster growth rate, $N_r$, but not $N_b$, remains accurate after several generations of exponential growth (Fig. S4B). Compared to $N_b$, $N_r$ is also more resistant to changes in the FP of the more diverse, slow-growing population (Fig. S4C) or the less diverse, fast-growing population (Fig. S4D). These results are consistent in libraries containing 1,000 or 10,000 barcodes. Furthermore, $N_s$ more often defines $N_r$ than max($N_b$). Together these simulations reveal that $N_r$ provides a more robust estimate of FP than $N_b$; in addition, they demonstrate that accuracy of $N_r$ at high FPs is greater in the 10,000-barcode library than in the 1,000-barcode library (Fig. S4C).

**Identifying, quantifying, and visualizing shared barcodes between samples.** Barcoded libraries also permit analysis of inter-organ dissemination by analyzing the similarity of tag frequencies between organs. Previous STAMP analyses identified the Cavalli-Sforza chord distance (33) between samples to quantify the genetic distance (GD), although other methods to assess allelic similarity between populations can be employed (20–22, 25, 26). GD is high when two samples are dissimilar, and low when they are more similar. We leveraged iterative barcode removal to obtain a more granular understanding of the similarity between samples. Our motivation arose from the fact that GD values are influenced by the abundance of tags in samples, as well as the number of shared tags. Highly similar populations (low GD values) can result from the sharing of many barcodes or very few highly abundant ones. Furthermore, the expansion of different clones that overlay similar populations yields high GD values (Fig. S5A), whereas the sharing of dominant clones between two samples yields low GD values, even if the underlying populations are dissimilar (Fig. S5B). We reasoned that additional metrics generated by our

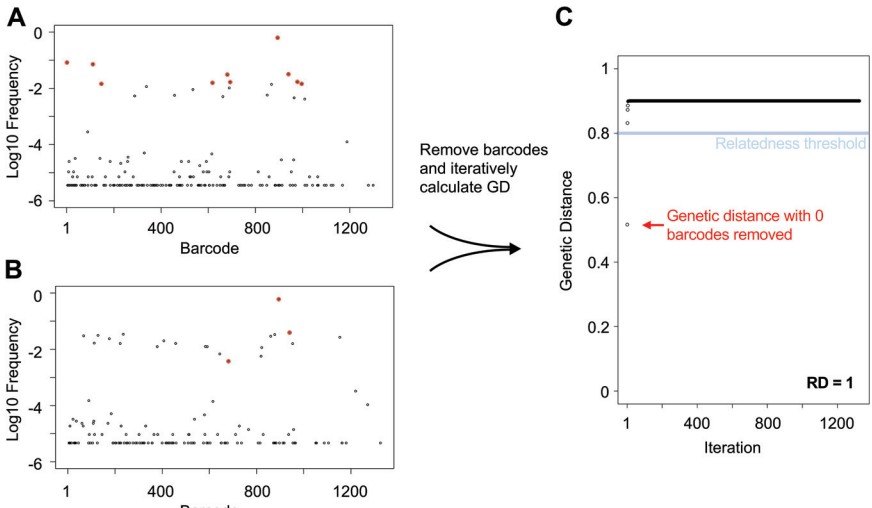

**FIG 3** Workflow for RD calculation. (A and B) Barcode frequency distributions in two samples. The top 10 most abundant barcodes in the top sample (A) are highlighted in red and identified in the bottom sample (B). The RD algorithm iteratively removes the most abundant barcodes and calculates GD after each iteration to generate the plot in C. These samples are moderately related to each other, since they share the same dominant barcode, but their underlying populations are dissimilar. RD is defined as the number of points on the plot in B that are below 0.8.

iterative barcode removal strategy could, when coupled with GD, help to characterize dissemination patterns more completely.

Similar to the approach taken with the resiliency algorithm for calculation of $N_r$, we iteratively removed the most abundant barcodes in both samples and created a score quantifying the number of barcodes that contribute to genetic similarity between the samples (RD, "resilient" genetic distance) (Fig. 3, Fig. S1). Low RD values indicate that the samples share relatively few barcodes. Samples with both low RD and low GD share only a few tags but are nevertheless highly similar; in this case, very few barcodes are shared, but they represent significant fractions of the total CFU in both samples (Fig. S5B). Samples with both high RD and GD share many tags, but the bulk of the population (in terms of CFU) are dissimilar; this can occur when different sets of bacteria expand in two samples, but both expansion events overlay relatively similar populations containing many barcodes (Fig. S5A). Samples with high RD and low GD are very similar and share many barcodes; this is typically observed in samples with high FP because they closely resemble the inoculum and therefore each other (e.g., early after infection) (Fig. S5C). Samples with low RD and high GD are completely dissimilar, suggesting they are unlikely to be related to each other either physically or temporally. Application of this approach to our ExPEC data proved valuable because it enabled us to distinguish between samples that were similar due to the dissemination of clones (low RD, low GD) versus when they were similar because they all closely resembled the inoculum (high RD, low GD, and high FP) (32).

Note that in this framework, "low" and "high" RD values are relative to the number of barcodes in the library. We created an additional metric where RD values are log-normalized (plus one) to the total number of detectable barcodes in the sample (an output from the resiliency algorithm). We refer to this metric as a fractional RD (FRD), which represents the relative abundance of shared barcodes in a pair of samples. FRD essentially normalizes RD across all samples, and therefore permits comparisons between samples. Similar to how high $N_r/N_b$ ratios signify the presence of expanded clones that overlay a diverse population, low FRD and low GD can signify the presence of abundant shared clones that overlay diverse dissimilar populations. The directionality of FRD calculations provides further information about similarity between

populations. For example, consider a situation where organ A and organ B are similar samples (low GD) that resemble the data in Fig. S5B. If RD = 11 (i.e., 11 barcodes are shared between A and B), $FRD_{A-B} = \log(11 + 1)/\log(B_B + 1)$, where $B_B$ is the number of barcodes in sample B, while $FRD_{B-A} = \log(11 + 1)/\log(B_A + 1)$, where $B_A$ is the number of barcodes in sample A. High $FRD_{A-B}$ signifies that the barcodes that were shared between the two populations represent a large fraction of all the barcodes in sample B. Correspondingly, a low $FRD_{B-A}$ means that the barcodes that are shared between A and B only represent a small fraction of the barcodes detected in sample A. The difference between $FRD_{A-B}$ and $FRD_{B-A}$ implies the existence of a larger, more diverse and dissimilar underlying population in sample A but not sample B. In this example, we can conclude that (i) samples A and B are similar (low GD); (ii) the similarity is driven by only a few barcodes (low RD); and (iii) these few barcodes represent a large population of sample B but overlay a more diverse resident population in sample A (high $FRD_{A-B}$, low $FRD_{B-A}$). Note that FRD is strictly a metric that uses the number of barcodes, not their abundance. Barcode abundance is considered in GD calculations, and therefore a combined approach using all of these metrics (GD, RD, and FRD) is superior to using any one metric individually.

**Reanalysis of *Pseudomonas aeruginosa* bacteremia.** We built our tools using a systemic model of ExPEC infection (32). We further tested these tools and associated metrics by reanalyzing data published in a recent study examining the trafficking of a barcoded library of *P. aeruginosa* following its intravenous inoculation into mice (24). This study revealed that gallbladder seeding by *P. aeruginosa* allows the pathogen to disseminate to the intestines and ultimately to be shed in the feces (24). Our reanalysis buttresses these conclusions and uncovered unappreciated patterns of *P. aeruginosa* expansion and dissemination that were hidden in the data sets due to additional variation in barcode frequencies not captured by $N_b$. In most of the samples, $N_r$ was greater than $N_b$ and, in many cases, the $N_r/N_b$ ratio was >10, particularly in the liver and lungs (Fig. 4A), indicating that there were significant clonal expansions at these sites. Here, clonal expansion refers to markedly uneven tag distribution in the sequencing data. Importantly, the $N_r/N_b$ ratio reveals the presence of clonal expansion, but not its biological source. For example, highly abundant barcodes could result from expansion that is confined to an organ or arise from transit from a different organ. This reanalysis also revealed marked heterogeneity in $N_r$ values within and between organs at 24 h postinfection, despite very similar $N_b$ values (Fig. 2C of reference 24). The large variance in $N_r$ values, for example in the liver and lung (Fig. 4A to C), reveals considerable differences in the sizes of the bottlenecks in different animals that were not captured by $N_b$. The barcode frequency distribution plots shown in Fig. 4B to G underscore that $N_b$ is extremely sensitive to highly abundant tags and therefore does not adequately capture and quantify the marked differences of pathogen population structure within and between organs. $N_b$ is more similar to $N_r$ when barcode frequencies are relatively even (compare Fig. 4D and E). Therefore, using $N_r$ in addition to $N_b$ enables a more complete understanding of the entire population structure in the host by accounting for less-abundant barcodes. These underlying populations are important to detect, as they may occupy distinct niches, contribute to persistent infections, or disseminate to other organs in the host.

The potency of our approach is well illustrated by reanalysis of the data from single animals infected with *P. aeruginosa*. For example, in mouse 1 (Fig. 5), there is a 2.5 log difference between $N_b$ and $N_r$ in the lung, suggesting a large clonal expansion. The lung sample was somewhat similar to the liver (GD = 0.66, RD = 739) and spleen (GD = 0.56, RD = 523), but completely dissimilar to intestinal organs (small intestine, cecum, colon, and feces) and the gallbladder (GD > 0.8, RD = 0) (Fig. 5), revealing that a set of dominant clones circulated systemically, but not enterically. However, the fact that these GD values are modest and not closer to 0 suggests that some dominant clones in each sample were not shared. Additionally, the relatively high RD values indicate that removal of a few dominant barcodes does not abolish genetic similarity.

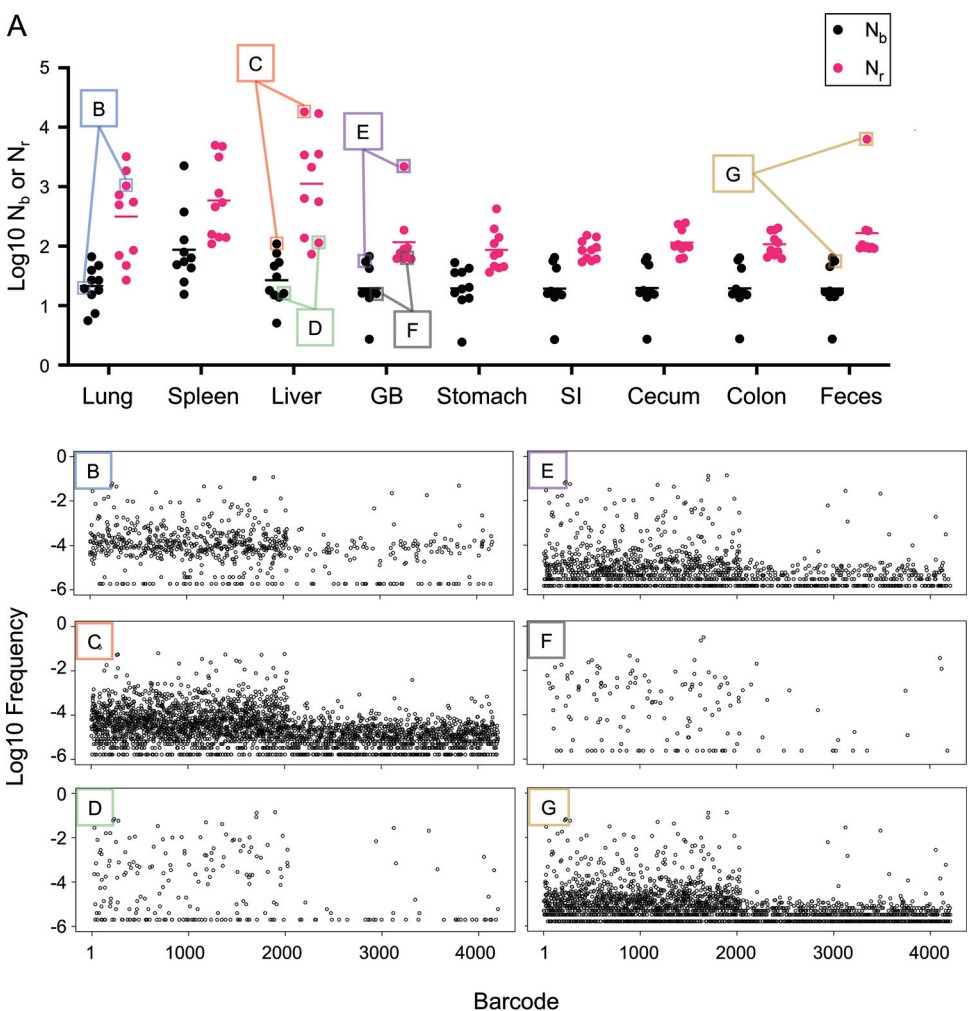

**FIG 4** Reanalysis of *P. aeruginosa* systemic infection population dynamics. (A) $N_b$ values are displayed across organs from Fig. 2C of reference 24, along with $N_r$ values determined here. (B to G) Barcode frequency distributions from individual samples B to G (as shown in A) are displayed to visualize the underlying distributions that give rise to $N_b$ and $N_r$ values. These distributions are prior to noise correction by the resiliency algorithm. These plots represent a wide range of barcode frequency distributions, even though $N_b$ is similar in all of them. $N_r$, by accounting for all barcodes, more robustly captures and quantifies the differences between these samples.

Therefore, the populations in these systemic samples consist of underlying subpopulations that are both similar and diverse. For example, the lung and liver do not share many highly abundant barcodes, but both samples have similar underlying populations (Fig. 5C, blue brackets). Comparisons of $N_r/N_b$ ratios in these organs also reveal that dominant clones are present in the lung, liver, and spleen (460, 165, and 24, respectively) (Fig. 5C). These observations are consistent with a model where the liver, spleen, and lung each received a large portion of the inoculum and had distinct clonal expansion events, some of which spread systemically. Elsewhere in the animal, there was marked sharing of barcodes between the gallbladder and the intestines (GD < 0.2) and these transferred barcodes comprised nearly all of the barcodes in the intestinal organs (FRD$_{gallbladder-intestine}$ > 0.9). Consistent with $N_r$ values, the small number of barcodes transferred between the gallbladder and liver (GD = 0.74, RD = 28) comprised a large fraction of the gallbladder population (FRD$_{liver-gallbladder}$ = 0.8) but only a small fraction of the liver barcodes (FRD$_{gallbladder-liver}$ = 0.42). These FRD differences reveal that large subpopulations of liver-resident bacteria are distinct from those in the gallbladder. Inspection of the barcode frequency distributions confirms that most expanded clones in the liver are not derived from the gallbladder (Fig. 5D), consistent with their

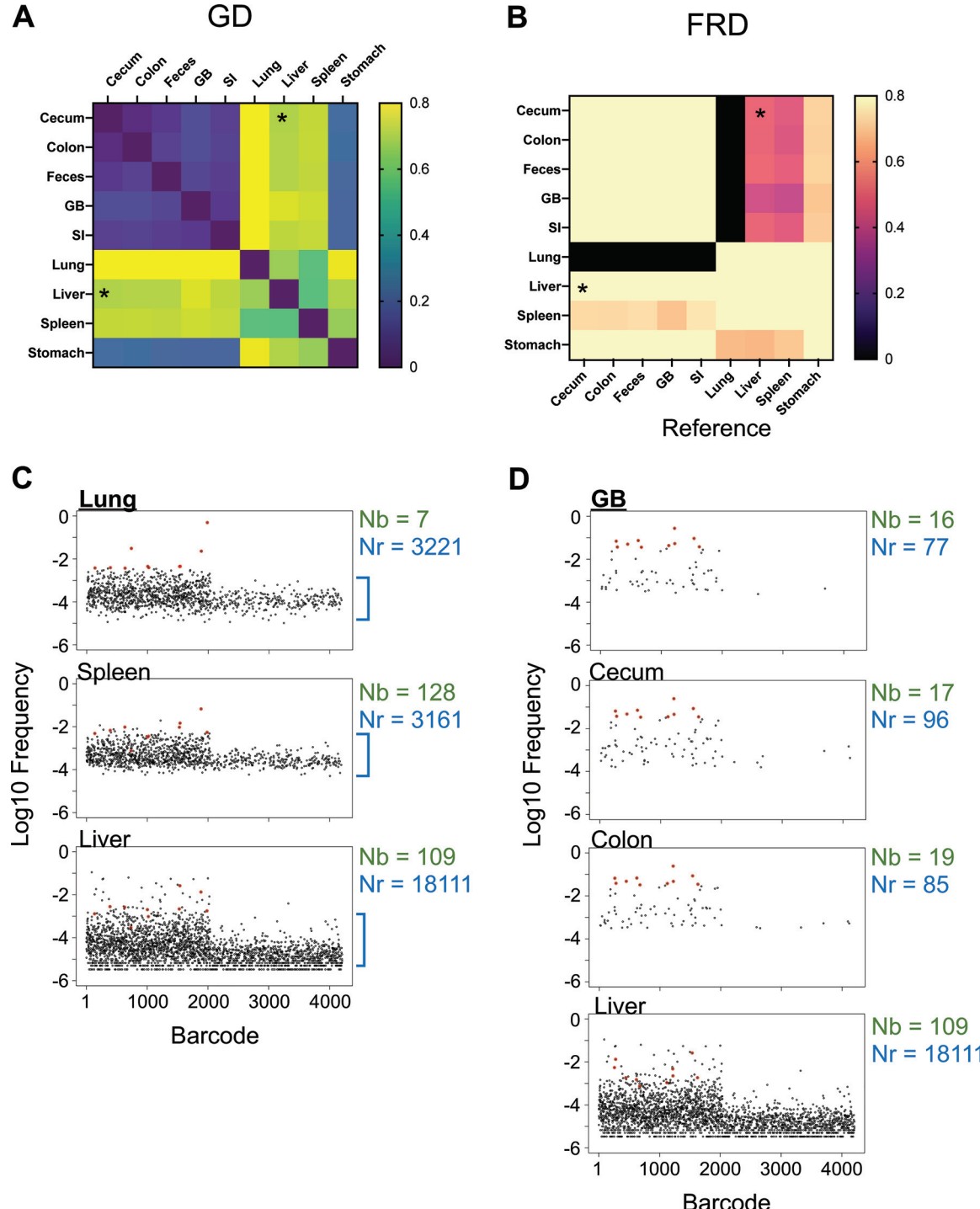

**FIG 5** Reanalysis of *P. aeruginosa* population dynamics in a single animal. (A and B) GD (A) and FRD (B) values were calculated for all organs in mouse 1 from Fig. 2C of reference 24. (A) Since GD is the same for a pair of samples in either direction, GD heatmaps are symmetric along the diagonal. (B) The asymmetry of color along the diagonal in the FRD heatmap arises from the fact that only one of the axes (the column names) serves as the reference, while the row names are simply the other sample in the pair used to calculate RD. The liver and cecum are modestly similar samples as measured by genetic distance; however, $FRD_{liver-cecum}$ is greater than $FRD_{cecum-liver}$ (asterisks). This indicates that the shared barcodes between the liver and cecum represent smaller fractions of the total liver barcodes than the total cecum barcodes. Therefore, the liver, but not the cecum, has a larger resident nonshared population. (C) Barcode frequency distributions after noise removal are shown for the lung. The top 10 barcodes are highlighted in red and identified in the spleen and liver samples, demonstrating that these samples share some, but not all, dominant tags. This is reflected in GD values in A. $N_r$ and $N_b$ values are displayed for reference. Blue brackets indicate the diverse underlying population. (D) Same as C but the gallbladder (GB) serves as the reference for the top 10 barcodes and these barcodes are identified in the colon, cecum, and liver.

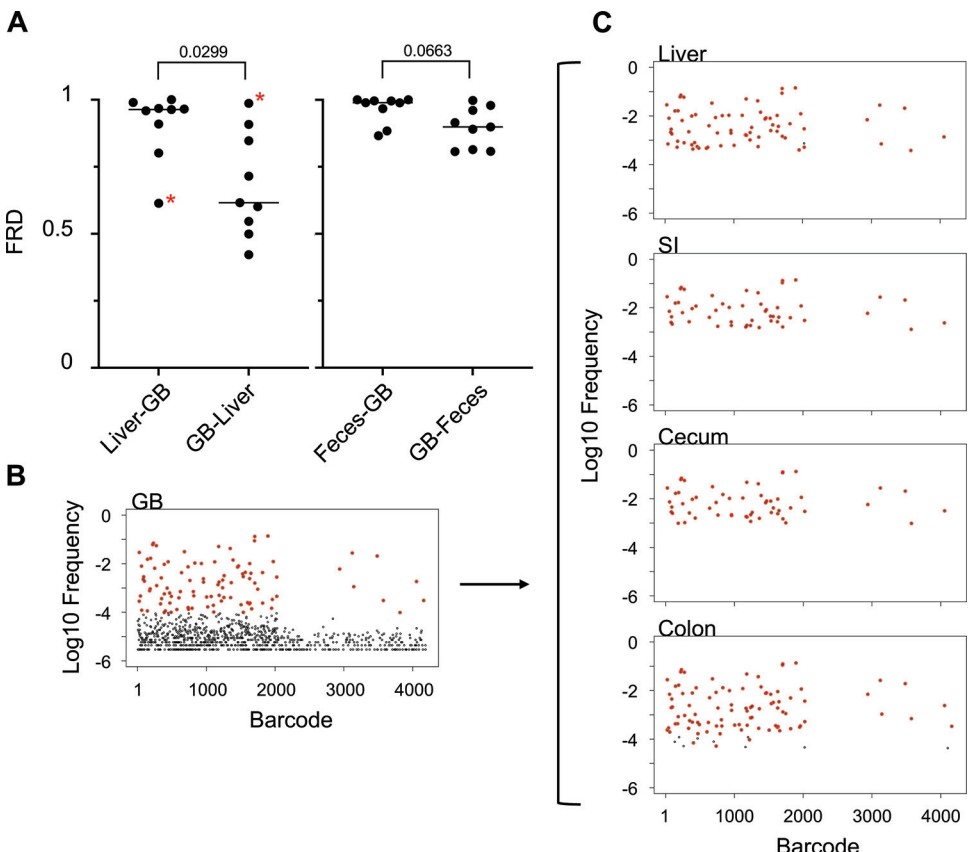

**FIG 6** Gallbladder transmission dynamics. (A) FRD values are displayed for liver/gallbladder and feces/gallbladder pairs. There is no significant difference between $FRD_{gallbladder-feces}$ and $FRD_{feces-gallbladder}$. In contrast, $FRD_{gallbladder-liver}$ is significantly less than $FRD_{liver-gallbladder}$ (two-tailed paired $t$ test), indicating that shared tags typically represent a smaller fraction of tags in the liver than the gallbladder. The difference in FRD values indicate that the liver has a resident population that is not shared with the gallbladder. Asterisks represent an animal (mouse 9) that was a notable exception to this trend. The barcode frequency distribution (after noise removal by the resiliency algorithm) of the gallbladder of this animal is presented in B. The top 100 most abundant barcodes in the gallbladder are highlighted in red, and these barcodes are highlighted in other organs in C. In this animal, the gallbladder appeared to be more diverse and only shared a fraction of its population with other organs.

expansion within the liver, independently of transit to/from the gallbladder. These analyses illustrate how our tools enable high-resolution mapping of population dynamics in a single animal.

In contrast to measuring multiple parameters in a single animal, comparing single metrics across animals enables detection of both consistent and heterogeneous facets of population dynamics. For example, $FRD_{liver-gallbladder}$ was significantly higher than $FRD_{gallbladder-liver}$, indicating that the bacteria that are shared between the liver and gallbladder consistently represented a larger fraction of the population in the gallbladder than in the liver (Fig. 6A). This contrasts to comparisons between the gallbladder and the feces, which have FRD values in both directions consistently near 1, suggesting that the fecal population is nearly entirely derived from the gallbladder (Fig. 6B). The underlying anatomy in this infection model likely explains these differences. As proposed by Batcha et al. (24), the liver first captures bacteria from blood and a small number of these cells then seed the gallbladder, where they subsequently replicate in bile. This model can explain why the liver often possesses its own resident population distinct from the gallbladder, and FRD enables robust quantification of this phenomenon. This pattern was observed in most animals, but mouse 9 was a clear exception (Fig. 6C). In this animal, the gallbladder population was one of the most diverse (high $N_r$) observed. Furthermore, the gallbladder population in mouse 9 was nearly identical to all other

organs (low GD). The gallbladder appears to account entirely for the population in the liver ($FRD_{gallbladder-Liver} = 1$, contrasting with mouse 1 in Fig. 5) and the gastrointestinal (GI) organs. The gallbladder population also includes a set of nontransferred barcodes that are absent from the liver and gastrointestinal organs ($FRD_{colon-gallbladder} = FRD_{cecum-gallbladder} = FRD_{SI-gallbladder} = 0.6$, while $FRD_{gallbladder-colon} \approx FRD_{gallbladder-cecum} \approx FRD_{gallbladder-SI} \approx 1$) (Fig. 6B and C). Thus, in mouse 9, the gallbladder seeded the intestines with only a fraction of its population.

By plotting the GD and FRD of each organ against all organs in the animal in a heatmap, we can rapidly detect consistent and variable spreading events. For example, examination of the heatmaps in Fig. 5 and Fig. S6 reveal two groups of animals that vary in the levels of systemic spread of gallbladder bacteria. In mice 1, 4, 5, 6, 7, and 10, the gallbladder appeared to be mostly dissimilar to the lungs and the spleen. In contrast, in mice 2, 3, 8, and 9, the gallbladder population was highly similar to that of the lungs and spleen. The variable magnitudes of the stochastic systemic spread of gallbladder bacteria, very likely via blood, potentially explain these distinct dissemination patterns. Therefore, the fate of bacteria that have seeded and replicated in bile can profoundly alter pathogen populations in distal organs. More broadly, the STAMPR framework developed here can rapidly uncover stochastic and more subtle patterns of dissemination.

**Summary and perspectives.** Coupling barcoded bacteria with high-throughput DNA sequencing enables powerful investigations of bacterial population dynamics. In infection biology, comparisons of barcode abundances in an experimental inoculum with those found in various host organs (the output) at different times postinoculation enables inferences about the sizes of bottlenecks and patterns of pathogen dissemination. However, bacteria often undergo unequal expansion within an organ, resulting in marked differences in the frequencies of barcodes in the output library compared to the input. Here, we show that these differences confound calculations underlying quantification of founding population sizes and dissemination patterns. We created a new framework, called STAMPR, that provides a more comprehensive assessment of within-host population dynamics.

Our approach accounts for unequal growth and highly abundant tags to provide a more complete assessment of infection population dynamics. Two metrics ($N_r$ and $N_b$) define the number of organisms from the inoculum that give rise to the population in an organ. $N_b$ is highly sensitive to disproportionately abundant tags, while $N_r$ is more resistant to the presence of highly abundant barcodes. The $N_r/N_b$ ratio measures the magnitude of unequal growth, which is often very large in samples where very few clones have expanded dramatically. Comparison of barcode frequencies between samples further enables assessment of bacterial dissemination, quantified by GD. We further refine GD to determine the precise number of barcodes that are transferred between samples, in a metric termed RD. Combining RD with founding population sizes results in a directional metric (FRD) that quantifies the relative abundance of shared bacteria within two samples. Taken together, we refine and establish eight metrics for a pair of samples ($N_b$ and $N_r$ for samples A and B, and GD, RD, $FRD_{A-B}$, and $FRD_{B-A}$) from which the entire underlying barcode frequency distributions can be summarized (Fig. S1). Furthermore, when used across many organs, these metrics enable high-resolution analysis of population dynamics in a single animal.

Reanalysis of previous infection data also highlights the power of our method to uncover previously unappreciated dissemination dynamics. Importantly, though our approach removes clonal expansions for more accurate calculation of the FP, it also identifies them. Analyses of these heterogenous expansion events in organs and across animals reveal that this previously unrecognized phenomenon is highly prevalent in infection contexts. Approaches to visualize and quantify such events will set the stage for future studies to characterize how host responses, spatial relationships, and interventions may govern these uneven replication dynamics. Future studies can provide further resolution by employing these metrics with repeated sampling over time, which would enable more precise determination of rates of population constriction

and growth. We anticipate that applications of our new tools in future studies will deepen our understanding of within-host (and between-host) bacterial population dynamics. Finally, our strategy to account for unequal tag abundance will also have utility in studies beyond infection dynamics that rely on barcode frequency analysis, including lineage tracing, cancer progression, and experimental evolution (18).

## MATERIALS AND METHODS

**Processing of STAMP reads.** Reads were first demultiplexed on Illumina BaseSpace via i7 and then further demultiplexed on CLC Genomics Workbench using the first 6 nucleotides. Trimming was performed using the default parameters and only reads between 18 and 22 nucleotides (nt) were kept. Trimmed reads were mapped to the list of 1,329 barcodes (obtained from reference 32) using the default parameters in CLC and the mapping file was exported directly from CLC as a csv file containing barcodes and read counts.

**Calculation of $N_b$ and $N_r$.** Previous studies have "calibrated" $N_b$ values to a known standard curve and the calibrated values were referred to as $N_b'$. However, this calibration is only meaningful when the biological data generally satisfies the assumptions of equal growth rates and uniform bottleneck that is used to generate the standard curve and was therefore omitted from the analyses presented here. Comparison of Fig. 5 in this study (uncalibrated) and Fig. S2C from reference 24 (calibrated) shows the negligible impact of calibration on these data. To calculate $N_b$ and $N_r$, metadata is first retrieved from a csv file containing the barcode frequencies of the references and samples and from a table of CFU for each sample. Replicates for reference vectors (i.e., values sequenced from the inoculum) are averaged. A bottleneck for the reference vector is then iteratively simulated by resampling the reference vector from a multivariate hypergeometric distribution. Each iteration is resampled to different depths, ranging from 1 read to 10 times the total number of barcodes in the sample in increments of 10. Therefore, a library with 1,000 barcodes is iteratively resampled 1,000 times from 1 read to 10,000 reads. This is typically sufficient to plateau the number of unique barcodes. At each iteration, the number of nonzero barcodes is calculated and plotted against the resampling size. This plot is referred to as the "reference resample plot." The $x$ axis value of this plot is referred to as $N_s$ and represents a bottleneck size that yields a desired number of barcodes. This plot is used later to identify the size of the computation-derived bottleneck ($N_s$) that gives rise to the observed number of barcodes in the sample.

A user-specified noise-filtering step is included to assist the resiliency algorithm in locating noise. In practice, this is estimated from control samples within a sequencing run for which the precise number of barcodes is known. Reads that map to other barcodes are therefore a result of noise, likely due to index hopping. Measuring the relative abundance of these reads enables a preliminary user-controlled noise filtering prior to the more unbiased steps in the algorithm, as described below. For the ExPEC study, noise was set to 0.5% (indicated by controls), while it was set to 1% for reanalysis of *P. aeruginosa* data (a conservative estimate). The desired number of reads is simulated on the reference vector with a multivariate hypergeometric distribution and subtracted from the output vector.

Next, we determine the number of required iterations for barcode removal, which is set to a minimum value among the CFU of the sample (plus one) or the number of barcodes with more than 1 read. This ultimately helps speed computation of $N_r$ for low CFU samples, since it is not necessary to iterate for more than the number of unique cells contributing to DNA in the sample. The reference and output vectors are then matched and ordered by the output vector, and the first $N_b$ is calculated from the Krimbas and Tsakas equation. Next, the last row that contains the most abundant output barcode is removed, along with the corresponding input barcode. Note that at early iterations, this is essentially the same as setting the output barcode equal to the input barcode. After this removal, the second $N_b$ is calculated; this process is then iterated for the previously determined number of iterations to generate the first resiliency plot.

A local minimum can arise in the resiliency plot when barcodes resulting from bacteria present in the sample ("real" barcodes) have been removed. This is due to the relatively similar sequence noise across all samples, which are multiplexed in >50 samples per MiSeq lane. As the real barcodes have been removed, the "noise" resembles the inoculum and begins to raise the $N_b$ value. For example, if there are 100 "real" evenly distributed barcodes in the sample and 100 "noise" barcodes, removal of the 90 most abundant barcodes will yield a population that resembles one where there is no noise, but 10 highly abundant clones overlaying a more diverse population. This results in a low $N_b$ value. When the 100 more abundant barcodes are removed, there are no longer any highly abundant barcodes, so $N_b$ increases. Biological data, however, is rarely this clear, and therefore a goal of the algorithm is to identify all local minima, as they represent potential locations in which the barcode distribution could be approaching noise.

To accomplish this, the algorithm starts at multiple "initiation sites" across the resiliency plot. The number of initiation sites is set equal to 1/15 the number of elements in the resiliency plot, which can be calibrated as needed but is practical for STAMP data with ∼$10^2$ to $10^3$ barcodes. Each initiation site is an $x$ coordinate on the resiliency plot. For each of these sites, the algorithm performs the following computation. A sample is drawn from a normal distribution with a mean equal to the position of the initiation site and standard deviation equation to 1/10 the number of elements in the resiliency plot. Decreasing the standard deviation decreases the "search space" and, therefore, increases the number of local minima that can potentially be found. If the standard deviation is too large, only the global minimum will be found. This sample is a "guess" for where to potentially move on the resiliency plot. Since this "guess" is another $x$ coordinate on the resiliency plot, the corresponding $y$ coordinate is determined. If this new $y$ coordinate is less than the $y$ coordinate of the initiation site, the mean of the next normal

distribution is set to equal the guess. This process is repeated 1,000 times, where "guesses" are repeatedly drawn from a normal distribution and accepted only if they result in a lower $y$ coordinate on the resiliency plot. In this manner, the initiation site settles to some value on the resiliency plot. The $x$ coordinates where this process settles after 1,000 iterations and across all initiation sites are known as "breaks." The location of the greatest log change in the resiliency plot is also determined and added to the breaks; a similar parameter was used to separate true and false barcodes in a previous approach (19). Collectively, each break represents some notable transition in the barcode frequencies of the output sample relative to the input.

An "indices table" is then constructed around the breaks. For each break, fractional abundance for those barcodes in between is calculated (referred to as "weights"). For example, if two breaks are located at position 5 and 200 in the resiliency plot, then we calculate the fractional abundance of the first 5 barcodes and barcodes 6 to 200. Additionally, we identify the maximum $N_b$ up to each break. In this example, this would mean identifying the maximum $N_b$ in the first 5 values of the resiliency plot and the maximum $N_b$ in the first 200 values. The indices table combines the maximum $N_b$, weight, and breaks.

Next, noise is defined from the indices table as the greatest log change in weight; the breakpoint immediately prior to the greatest log change represents the iteration in the resiliency plot after which all real barcodes have been removed. Since the resiliency plot is derived from an ordered list of barcode frequencies, this iteration can be traced back to barcodes above and below a specific number of reads. A verification step is performed to ensure that all non-noise barcodes represent a set minimum of the total number of reads (97% in this study) and can be altered as needed. After noise is determined, all barcodes determined to be noise are set to 0 and a new resiliency plot is generated from this noiseless set of data.

The final FP estimation from the resiliency algorithm, referred to as $N_r$, is equal to the maximum value among (i) the maximum $N_b$ in this new second resiliency plot, (ii) the original $N_b$ estimate (i.e., the output of the first iteration), or (iii) the value of $N_s$ which corresponds to the number of non-noise-derived barcodes derived from the reference resample plot using inverse interpolation. This ensures that $N_r$ will always be greater than or equal to $N_b$ and that $N_r$ will never be less than the observed number of counted barcodes. In this manner, this algorithm chooses the strategy that determines the FP that most adequately captures all barcodes. Very complex libraries (e.g., >100,000 barcodes) would almost always derive $N_r$ values from the reference resample plot, while smaller libraries will more often use resiliency plots to find the maximum $N_b$ for large FPs. Similarly, if there is a substantial amount of variation in barcode frequency due to sources other than the bottleneck (such as phenotypic heterogeneity) such that $N_b$ is very low, $N_r$ will be equal to $N_s$. An important implication of the use of $N_s$ by the resiliency algorithm is that the resolution limit of $N_s$ increases when there are more barcodes (Fig. S4B). For example, if the data is highly variable but all barcodes are present, $N_s$ will be greater for a library of 1,000 barcodes than for a library of 200 barcodes. Additionally, since $N_r$ relies on simulations, the precise value differs slightly each time the algorithm is run. One notable class of edge cases is samples with 1 CFU that can yield $N_r$ values of ~2; these cases can easily be corrected *post hoc* and do not affect data interpretation.

**Calculation of GD and RD.** Genetic distance (GD) is calculated by the Cavalli-Sforza chord distance as described (22, 33). We analogize our approach for calculating $N_r$ to genetic distance and created a metric—RD—that measures the number of barcodes that contribute to "meaningful" relatedness between samples. Low values of RD imply that few barcodes are shared, whereas high values of RD imply that many barcodes are shared.

RD is calculated in single script as follows. Barcode frequency vectors are obtained after running the resiliency algorithm after removal of noise. Both organs are paired and ordered by the geometric mean abundance of each barcode. GD is calculated iteratively and barcodes are removed as done in the resiliency algorithm. RD is equal to the number of barcodes that yield GD values below 0.8 on the graph of GD versus iteration. Figure S5 shows how this graph behaves for a variety of given inputs and how the RD value is derived from them. The value 0.8 approximates the GD of two unrelated biological samples (24), but this threshold can be adjusted depending on how the experimenter interprets "meaningful" relatedness. In Bachta et al., this threshold was determined by calculating inter-animal GD, where these samples are expected to be completely dissimilar. To assess the validity of this threshold without animals, we simulated a pair of random samples with varying FP values and calculated GD (Fig. S7). The resulting curve reveals that two random samples with higher FP values will also have lower GD values, since the odds of the same barcodes being present in a pair of samples increases with higher FPs. In both ExPEC and *P. aeruginosa* libraries, GD = 0.8 intersects the curves after the upper asymptote but before the steep decrease in the sigmoid. By plotting $\log_{10}$(FP) versus GD, future studies can verify that GD = 0.8 intersects this curve at a similar location.

FRD is manually determined by dividing the log of each RD value (plus one) in each column of the output table (all pairwise comparisons) by the log of the maximum value in each column (plus one). The maximum value of each column is the RD value of the sample compared with itself, which defines the column.

**Data and code availability.** All scripts used in this manuscript are available at https://github.com/hullahalli/stampr_rtisan. Barcode frequency counts for ExPEC STAMP experiments were experimentally derived from our companion manuscript (32) and are available at the above link to reproduce plots in Fig. 2 and 3, and Fig. S5. Barcode counts for *P. aeruginosa* STAMP experiments are provided in reference 24.

## SUPPLEMENTAL MATERIAL

Supplemental material is available online only.

**TEXT S1**, DOCX file, 0.02 MB.

**mSystems**®

**FIG S1**, PDF file, 0.04 MB.
**FIG S2**, PDF file, 0.04 MB.
**FIG S3**, PDF file, 0.03 MB.
**FIG S4**, PDF file, 0.6 MB.
**FIG S5**, PDF file, 1.1 MB.
**FIG S6**, PDF file, 0.4 MB.
**FIG S7**, PDF file, 0.1 MB.
**TABLE S1**, DOCX file, 0.01 MB.

## ACKNOWLEDGMENTS

We thank members of the Waldor lab for providing valuable feedback on the manuscript. We are especially grateful to Michael Chao, Gabriel Billings, Brandon Sit, Ian Campbell, Sören Abel, and Pia Abel zur Wiesch for feedback on the manuscript.

This work was supported by an NSF Graduate Research Fellowship (K.H.), the Howard Hughes Medical Institute (M.K.W.), and AI-RO1-042347 (M.K.W.).

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
