## [Reviewer comments · mSystems]

Refined quantification of infection bottlenecks and pathogen dissemination with STAMPR

Karthik Hullahalli, Justin Pritchard, and Matthew Waldor

Corresponding Author(s): Matthew Waldor, Harvard Medical School

Review Timeline:

Submission Date:

July 8, 2021

Accepted:

July 30, 2021

Editor: Nandita Garud

Reviewer(s): The reviewers have opted to remain anonymous.

Transaction Report:

DOI: <https://doi.org/10.1128/mSystems.00887-21>

July 30, 2021

Dr. Matthew K. Waldor
Harvard Medical School
Boston

Re: mSystems00887-21 (Refined quantification of infection bottlenecks and pathogen dissemination with STAMPR)

Dear Dr. Matthew K. Waldor:

Your manuscript has been accepted, and I am forwarding it to the ASM Journals Department for publication. For your reference, ASM Journals' address is given below. Before it can be scheduled for publication, your manuscript will be checked by the mSystems senior production editor, Ellie Ghatineh, to make sure that all elements meet the technical requirements for publication. She will contact you if anything needs to be revised before copyediting and production can begin. Otherwise, you will be notified when your proofs are ready to be viewed.

As an open-access publication, mSystems receives no financial support from paid subscriptions and depends on authors' prompt payment of publication fees as soon as their articles are accepted. =

Publication Fees:

We recognize that the video files can become quite large, and so to avoid quality loss ASM